# Carbon-Nanotube-Based Nanocomposites in Environmental Remediation: An Overview of Typologies and Applications and an Analysis of Their Paradoxical Double-Sided Effects

**DOI:** 10.3390/jox15030076

**Published:** 2025-05-21

**Authors:** Silvana Alfei, Guendalina Zuccari

**Affiliations:** 1Department of Pharmacy (DIFAR), University of Genoa, Viale Cembrano, 4, 16148 Genoa, Italy; 2Laboratory of Experimental Therapies in Oncology, IRCCS Istituto Giannina Gaslini, Via G. Gaslini 5, 16147 Genoa, Italy

**Keywords:** environmental pollution, environmental remediation, carbon nanotubes (CNTs), functionalized CNTs, water and air detoxification, water desalination, CNT-based membranes, CNT regeneration, CNT toxicity

## Abstract

Incessant urbanization and industrialization have resulted in several pollutants being increasingly produced and continuously discharged into the environment, altering its equilibrium, with a high risk for living organisms’ health. To restore it, new advanced materials for remediating gas streams, polluted soil, water, wastewater, groundwater and industrial waste are continually explored. Carbon-based nanomaterials (CNMs), including quantum dots, nanotubes, fullerenes and graphene, have displayed outstanding effectiveness in the decontamination of the environment by several processes. Carbon nanotubes (CNTs), due to their nonpareil characteristics and architecture, when included in absorbents, filter membranes, gas sensors, etc., have significantly improved the efficiency of these technologies in detecting and/or removing inorganic, organic and gaseous xenobiotics and pathogens from air, soil and aqueous matrices. Moreover, CNT-based membranes have displayed significant potential for efficient, fast and low-energy water desalination. However, despite CNTs serving as very potent instruments for environmental detoxification, their extensive utilization could, paradoxically, be highly noxious to the environment and, therefore, humans, due to their toxicity. The functionalization of CNTs (F-CNTs), in addition to further enhancing their absorption capacity and selectivity, has increased their hydrophilicity, thus minimizing their toxicity and carcinogenic effects. In this scenario, this review aims to provide evidence of both the enormous potential of CNTs in sustainable environmental remediation and the concerning hazards to the environment and living organisms that could derive from their extensive and uncontrolled utilization. To this end, an introduction to CNTs, including their eco-friendly production from biomass, is first reported. Several literature reports on CNTs’ possible utilization for environmental remediation, their potential toxicity due to environmental accumulation and the challenges of their regeneration are provided using several reader-friendly tools, to better capture readers’ attention and make reading easier.

## 1. What Are Carbon Nanotubes (CNTs)?

Carbon nanotubes (CNTs) are nanomaterials made basically of carbon and were discovered for the first time by Sumio Iijima in the year 1991. They appear as thin and long cylinders [1], and their main physicochemical characteristics, most important structural aspects and most used methods for synthesizing them have been recently reported [2,3].

CNTs possess several thermal, electronic, mechanical and structural properties, depending on their different existing forms [3]. The different CNTs mainly present diverse lengths, diameters, chirality or rotational aspects [3]. Many possible applications of CNTs already exist, such as their use in solar panels, conducting and/or waterproof paper, supports for catalysis, nanoporous filters, coatings and the I/R optics industry. CNTs can, therefore, be considered the most relevant and valuable nanomaterial used until now; however, more knowledge of their nanotoxicology is urgently needed for a more secure production on a large scale and extensive employment.

### 1.1. Possible Structures of CNTs

During the formation of carbon-based materials, pure carbon can be organized in different architectures. Among them, pure carbon can be organized in the typical cylindrical structures of CNTs, including mainly single-walled carbon nanotubes (SWCNTs), double-walled carbon nanotubes (DWCNTs) and multi-walled carbon nanotubes (MWCNTs). SWCNTs derive from the rolling up of a thick sheet of graphite (graphene) [3,4] (Figure 1).

SWCNTs exhibit important electric properties that are not shared by other diversely structured CNTs, such as MWCNTs. SWCNTs are promising for scaling down electronics far below micro-electromechanical dimensions, which is the current challenge of modern electronics. The most basic building block of these systems is electric wire, and SWCNTs can be excellent conductors. One useful application of SWCNTs is in the development of the first intramolecular field effect transistors (FETs).

Depending on how graphene is packaged into the CNT tube, different possible configurations can be assumed by SWCNTs, including armchair, zigzag and chiral configurations (Figure 2).

The structural conformation of SWCNTs strongly influences their mechanical, electrical and thermal conductivity properties [3,5,6,7,8,9,10]. Therefore, the possible correlation between the structure assumed by SWCNTs and their mechanical, thermal and/or electrical properties has been an object of extensive study over the last few decades. Conversely, the structure of MWCNTs consists of more sheets of graphite wrapped around each other to construct a cylindrical shape. MWCNTs can assume the Russian Doll structural model, in which concentric SWCNT cylinders form the structure of MWCNTs (Figure 3a). Otherwise, the Parchment structural model consists of a single layer of graphite rolled in around itself, resembling the aspect of a rolled up newspaper (Figure 3b).

The external cylinders of MWCNTs can protect the inner ones from undesired interactions with exterior chemicals, thus preventing possible unwanted degradations. Double-walled carbon nanotubes (DWCNTs) are also known, in which only two SWCNTs are disposed, one inside the other (Figure 3c).

### 1.2. Main Physicochemical Properties of CNTs

The atomic structure of CNTs is very stable, with SWCNTs being more stable than MWCNTs [13]. While no dimensional limit is given for the length of a CNT, which is usually remarkably greater than the measure of their diameter, limitations exist for the size of their diameter, which should be <100 nm [3]. Table 1, reproduced from Aslam et al. [14], reports the main physical characteristics of SWCNTs and MWCNTs.

Post-synthesis modifications of CNTs, carried out to enhance their dispersity in water, their solubility in general and their physicochemical properties, while reducing their toxicity, are very common. Table 2 reports some structural properties demonstrated by several modified (M) and/or activated (A) CNTs, compared with those of unmodified ones (first row).

### 1.3. Methods for Synthesizing CNTs

The production of SWCNTs is more challenging than that of MWCNTs, and the properties and behavior of MWCNTs can differ significantly from those of SWCNTs. The application of SWCNTs to obtain CNT-improved nanocomposites is particularly laborious due to their problematic dispersion. Conversely, the dispersion of MWCNTs is simpler. Regardless, it is widely reported that SWCNTs possess features superior to those of MWCNTs. Therefore, scientists are mainly focused on finding more practical, lower-cost and sustainable modes of producing SWCNTs on a large scale than on MWCNTs. Appendix A collects the most utilized procedures for preparing SWCNTs and MWCNTs. Additionally, CNTs can naturally form in flames regularly emitted when methane, ethylene and benzene are burned, as well as in smoking in both indoor and outdoor air [23]. Unfortunately, as-produced CNT-based structures are affected by highly irregular dimensions and low-quality defective structures due to the uncontrolled conditions of production. Lacking the high degree of uniformity required by both research and industry, such naturally formed CNTs are more hazardous to the environment and living beings and cannot find practical application. Regardless, continuous efforts are focusing on trying to control environmental flames by theoretical models to produce more valuable CNTs on a large scale, at low cost, avoiding the use of risky chemicals and catalysts at the same time [24,25,26,27,28]. Currently, among the most traditional methods, including arc discharge [29], laser ablation [30] and chemical vapor deposition, chemical vapor deposition is the most performant and extensively utilized procedure for producing large-scale CNTs [31]. Unfortunately, this method requires CH_4_ and C_2_H_2_, which derive from coal and petroleum raw materials, as a source of carbon, as well as catalysts, such as Fe, Co and Ni NPs. In this regard, the major limitations of CVD processes consist in the necessity to utilize non-renewable carbon supplies and in the great emission of waste gases, such as toluene [32], thus worsening energy shortages and environmental pollution [33]. To address these issues, new eco-friendly, safer, low-cost and renewable raw materials as a source of carbon are urgently needed to prepare CNTs, and a more extensive study to find new methods for preparing CNTs is necessary.

#### Environmentally Friendly Synthesis of CNTs: Utilization of Biomass Raw Materials

Biomass, available in several forms, is distributed worldwide with abundant disposal. Biomass contains neutral carbon and represents a unique renewable energy source containing carbon. Moreover, biomass can provide gases, liquids, solid fuels and chemical raw materials. Biomass is of paramount utility in limiting global climate change, in overcoming the gap between energy supply and demand, in protecting the ecological environment and in realizing carbon neutrality [34]. The term biomass indicates any form of waste material deriving from animals and plants. Lignocellulosic and non-lignocellulosic biomasses are the most recognized classes of biomass [35]. The first class comprises woody biomass and herbaceous crops, including straw, rice husks, bean straw and sawdust, and is mainly made of cellulose, hemicellulose and lignin [36,37]. The second class embraces human and animal waste and microorganisms and their derivatives, and it mainly contains lipids, carbohydrates and proteins [38,39]. Carbon-based materials containing the glucose-based linear polymer cellulose, the heterogeneous polysaccharide hemicellulose and the three-dimensional phenylpropanoid compound lignin [40,41] are generally obtained using lignocellulosic biomass. The “Solid–Solid–Solid” and the “Solid–Gas–Solid” two-step methods are the most common methods used to obtain CNTs from biomass. Additionally, a one-step microwave pyrolysis method also exists [33]. Table 3 reports some examples of CNT production using different types of lignocellulosic biomass as carbon sources or for other functions.

As shown in Table 3, inexpensive and green renewable biomass can simultaneously act as a source of carbon, a catalyst and a support for a catalyst, which are essential for the efficient synthesis of CNTs. Appendix A reports several other relevant studies on the preparation of CNTs using different types of biomass as sources of carbon. Pyrolysis conditions and the physicochemical properties of products are also reported in Appendix A. Although traditional methods for pyrolyzing biomass provided good results, using volumetric microwave pyrolysis, an inside-out heating capable of deeply penetrating into the biomass particles occurs. The more rapid decomposition of biomass generates more biogas and a solid carbon source for synthesizing CNTs [46]. In addition to generating gas and solids in higher yields than traditional methods, biomass microwave pyrolysis lowers the yields of liquid. Generally, the typical gas, liquid and solid yields derived from microwave pyrolysis are around 46 wt% [46]. Due to the distinctive benefits deriving from the uniform, fast and volumetric heating offered by microwave-assisted pyrolysis, CNTs prepared from different biomasses can form and grow at lower temperatures than during heating by conventional methods. The solid carbon material (biochar, BC) achieved via microwave pyrolysis demonstrated better pore structures and, simultaneously, functioned as a good catalyst support for CNT synthesis. In the microwave electromagnetic field, the gas produced by biomass pyrolysis is ionized to create a gas discharge plasma (GDCP). The GDCP, associated with robust thermal radiation and heat conduction, allows an increase in the average temperature of BC, identified as the hot spot effect (HSE). In turn, the HSE generates a local high temperature that, following reduction of the carbon source gas, triggers the catalyzed growth of CNTs. In summary, microwave pyrolysis is suggested when a biomass as a source of carbon, a metal catalyst and a support are required for CNT preparation [46]. However, due to the numerous compositions and structures of biomasses, important factors and parameters for optimizing the process, as well as the mechanisms governing the preparation of CNTs by microwave pyrolysis, are still unclear. Appendix A comprises important information on CNTs obtained using biomass pyrolysis by microwave heating. Moreover, CNT-based nanocomposites can be prepared by creating junctions between two or more sorts of CNTs [47,48] or between them and graphene [49,50]. Connections between CNTs were found in CNTs synthesized using arc discharge (AD) or chemical vapor deposition (CVD) procedures. Lambin et al. theoretically studied the electronic properties of such junctions for the first time [51]. The so-called pillared graphene is just based on CNT–graphene junctions. It is characterized by three-dimensional (3D) carbon nanotube (3D-CNTs) structures and was experimented with as a building block for engineering 3D macroscopic architectures [52]. Such CNT-improved nanocomposites, encompassing only carbon, having porous scaffolds, macro-, micro-, and nanopores, and tunable pores, can find several applications in various sectors. Next-generation energy storage systems, supercapacitors, field emission transistors, high-performance catalysts, photovoltaics, and biomedical devices, such as implants and biosensors, can be developed using pillared-graphene-based nanocomposites [53,54,55].

## 2. CNTs for the Removal of Environmental Xenobiotics

The large number of pollutants continuously discharged into the environment, due to persistent urbanization and industrialization, has impaired the Earth’s natural equilibrium, reducing the environment to a risky state. Clean air, water, and soil have become rare, thus raising a worldwide concern for the health of humans, animals, plants and water-living organisms [56,57,58]. In this scenario, restoring environmental health is urgent and of paramount importance. Research should be extensively concentrated on finding new and advanced materials for efficient detoxification of gas streams, contaminated soil, water, wastewater, groundwater and industrial waste [56,59,60,61]. In this regard, nanotechnology has demonstrated high performance, thus finding applications in several sectors. Improved engineering and bioengineering devices, electronic and optoelectronic items, and medicinal and pharmaceutical tools have been developed using nanotechnology [62,63,64]. Nowadays, nanotechnology is attracting researchers as a discipline possessing great potential for providing effective options for environmental detoxification. In summary, nanotechnology is very promising for providing both safer and sustainable renewable energy sources, and for the development of new efficient strategies for water, soil and air decontamination [65]. Several carbon-based nanomaterials (CNMs), such as fullerenes, quantum dots, nanotubes, graphene and bulk carbon powder, have displayed remarkable efficacy in environmental detoxification via several processes [62,66]. Using CNMs, filters membranes and other nanocomposites have been developed to adsorb harmful water pollutants [67] and to neutralize water and waterborne pathogens, thus preventing severe infections [68]. Such membranes have been experimented with transforming several dangerous substances polluting water into safer compounds, thus helping in environmental detoxification and in limiting these substances’ risky effects on ecosystems and humans [69]. Their efficiency in environmental remediation has been improved by superior synthesis, providing more typologies of CNMs variable in shapes and sizes, and by post-synthesis chemical modifications, with active constituents, thus further enhancing their power in addressing environmental challenges. Mainly, carbon nanotubes have demonstrated great potential in the field of sustainable environmental remediation. It has been reported that CNTs’ capacity for the adsorption of organic pollutants (OPs) is significantly influenced by the strong π–π interactions they can establish with OPs. However, they have demonstrated significant cytotoxicity in both in vitro and in vivo experiments due to their accumulation in the environment and long persistence within the body of living organisms [70,71]. In this context, CNT functionalization has been shown to be effective in enhancing their selectivity for several dangerous compounds, increasing their hydrophilicity and minimizing their carcinogenic potential [57,66,71,72]. Functionalized CNTs (F-CNTs) have been successfully used to remove air pollutants, as well as to realize water decontamination, soil amendment, wastewater treatment and industry waste management [72,73]. Recently, research has been mainly focused on the adoption of biomass-derived CNTs, as described above, for cleaning up the environment and removing pollutants [74,75,76]. The presence of CNTs in materials used for environmental remediation significantly enhances the removal of both inorganic and organic xenobiotics, as well as that of pathogens [77,78,79]. CNT-based nanocomposites have afforded highly sensitive gas sensors, characterized by the capability of fast responses to gas pollutants, with high efficiency in air pollution monitoring [80,81,82,83,84]. In the following sub-sections, the uses of CNTs for environmental remediation by different processes, including absorption, catalytic and photocatalytic degradation, and filtration, are discussed, in terms of water and air detoxification. The manufacturing of CNT-based sensors for air pollutant detection is also reported.

### 2.1. CNTs for Water Detoxification

Good-quality water sources are currently poor because of their irregular distribution in terrains and their pollution caused by incessant population growth and increasing industrial production with related wastes [85]. Carbon-based nanomaterials, mainly CNTs, have demonstrated remarkable efficiency in the detoxification of water from different pollutants, including heavy metals and inorganic and organic toxic compounds, as well as pathogen microorganisms, via adsorption, catalysis, separation and disinfection. Additionally, CNTs can generally function as safe biosensor electrode materials, while CNT-based membranes have been proposed as high-flow desalination systems [85]. However, reliable approaches and criteria for assessing risks and repercussions on the environment and living beings, which paradoxically can derive from the extensive application and exposure to CNTs, are still very limited [2]. Additionally, the effective regulatory mechanisms that are necessary to correctly address and discuss the possible negative effects of CNTs, thus allowing their safer production and use, are still in their infancy [2]. The major source of potable water for several nations and up to 2 billion individuals is groundwater (GW), which is of paramount importance for agricultural, mining and industrial activities [86]. More specifically, GW offers various services and benefits to societies, depending on its geographically varying properties. These services include the following:*Provisioning benefits*, intended as the withdrawal of GW for human water use purposes, including agriculture, industry and human settlements.*Regulatory assistance*, intended as the buffer capacity of GW, capable of regulating GW systems’ quantity and quality.*Supporting services*, on which GW-dependent ecosystems (GWDEs) and other GW-related environmental features rely.*Cultural assistance*, linked to leisure activities, tradition, religion or spiritual values, which are associated with specific sites.

However, GW is largely exposed to pollution by different xenobiotics, mainly deriving from industrial activities in continuous expansion. In this context, among the several sustainable remediation technologies experimented with, CNTs have demonstrated excellent efficiency in adsorbing contaminants present in GW. The excellent performance of CNTs in the detoxification of GW mainly derives from their robust mechanical strength, high stability and persistent endurance. These characteristics ensure an extended lifespan and allow a reduced number of replacements. Additionally, CNTs can be easily combined with the recognized technologies currently used to remediate GW. Such traditional procedures generally exploit permeable reactive barriers or filtration systems, whose overall efficiency in xenobiotic removal can be significantly improved by adding CNTs. Liang et al. simultaneously removed trichloroethylene (TCE) and Cd (II) from GW using Al_2_O_3_/MWCNTs nanocomposites, at pH = 7, reaching a maximum absorption capacity of 27.21 mg/g for TCE and 19.84 mg/g for Cd (II) [87,88]. Later, Jha et al. studied MWCNTs for removing TCE alone from GW [89]. By the incorporation of CNTs into polyvinylidene difluoride (PVDF) hollow fiber membranes, the authors improved their adsorption efficiency by 27% at 30 °C. Moreover, Desouky used MWCNTs modified with ethylenediamine tetra-acetic acid (EDTA) to absorb Fe (II) from GW, with a rate of adsorption (ROA) depending on pH, based on ion exchange and surface complexation mechanisms. Lico et al. realized the unleaded gasoline removal (up to 99.5%) from GW in 5–10 min, whose percentage depended on the amount of non-purified MWCNTs used, which were tested in the range of 0.2–0.8 g, with 0.7 being the optimum quantity [90]. By employing MWCNTs, Mpouras et al. removed Cr (IV) from GW, reaching the highest adsorption of 450 mg kg^−1^ working at a pH up to 6.3, and reaching the maximum removal efficiency (RE%) of 88% utilizing 50 g L^−1^ adsorbent [91]. Unconfined Fe_3_C-loaded N-doped CNTs composites prepared by Liu et al. demonstrated high efficiency in adsorbing and reducing Cr (VI, and showed higher performance compared to confined Fe_3_C-loaded CNTs in terms of reduction, adsorption and reusability. A removal equal to 40.8% (23.7 mg/g) of Cr (IV) at pH = 5.4 was observed [92]. Recently, Ye et al. intercalated tannic acid–Fe (III) complexes with the functionalized CNTs, achieving nanofiltration membranes that demonstrated an energy-efficient 80.5% selective removal of Ca^2+^ ions from GW, a high water recovery rate of about 156% and a notable 97.3% rejection rate for perfluorooctanoic acid (PFOA) [93]. In the context of water pollution, wastewater (WW), deriving from domestic, industrial and agricultural assets, represents a noteworthy fount of environmental contaminants. WWs, especially if not correctly treated, can contain high concentrations of inorganic and organic xenobiotics, as well as pathogens. Dyes, pesticides, pharmaceuticals and antibiotics among organic molecules, and heavy metals and radio nuclides among inorganic contaminants, can possess high toxicity levels, carcinogenic properties and resistance to environmental degradation. In addition, waterborne and non-waterborne pathogens and their toxins represent a high hazard for human and animal health, being capable of causing and sustaining severe infections, often resistant to antibiotic treatments, with a high mortality rate and a worrying economic impact. The World Health Organization (WHO) reported that more than 1.2 billion people have suffered illnesses or fatalities due to the consumption of polluted water and expects that this number will increase further due to enduring industrialization and urbanization [69]. As in the case of GW, among the several methods that have been proposed for the purification and treatment of WW [94,95,96,97,98], CNTs appear as the most promising material for WW remediation by adsorption, catalytic and photocatalytic degradation, separation and disinfection. This potential is mainly due to their ultrahigh water flow rates [99,100], outstanding sorption efficiency, and large surface area, which also offers several functionalization and modification opportunities [101]. Due to incessant research, also devoted to finding optimal work settings, several CNT-based adsorbents capable of effectively removing contaminants present in WW have been developed.

#### 2.1.1. CNTs for Decontamination of Water with Inorganic Xenobiotics via Different Absorption Mechanisms

Heavy metals are persistent environmental contaminants possessing high toxicity and carcinogenic effects, thus representing a high risk to the health of living beings, mainly due to their environmental accumulation. CNTs have been intensively experimented with as adsorbent materials for remediating heavy metals from water by different mechanisms. It has been demonstrated that their efficiency can be affected by several factors, mainly including the radius, electronegativity and the reduction potential of heavy metal ions, as well as the first stability constant of the associated metal hydroxides [102]. CNTs have also been studied for the removal of radionucleotides. Wang et al. adsorbed ^243^Am (III) on the surface of MWCNTs [103]. Collectively, CNTs interact with heavy metals mainly via electrostatic, ionic and chemical interactions between metals and the functional groups present on their surface [104]. These and other interactions, their mechanisms and the optimized operational conditions for the efficient removal of several inorganic pollutants from water are reported in Appendix A. It collects a series of 142 case studies on the utilization of CNTs to absorb various heavy metals. The subsequent Appendix A instead collects some studies about the use of pristine and/or modified CNTs to absorb mixtures of heavy metals from WW. In Appendix A, in addition to the main optimized operational parameters, CNTs’ removal efficiency (%) or absorption capacity (mg/g) and synthetic method are evidenced (second column).

#### 2.1.2. CNTs for Organic Xenobiotic Removal by Absorption

The concentration of organic xenobiotics of different types in WW has strongly increased in the last few decades due to high urbanization and augmented industrial production. Most organic xenobiotics are highly dangerous to living organisms due to their cytotoxic, genotoxic and cancerogenic effects. Additionally, they display low reactivity at low concentrations, thus rendering their removal from water extremely difficult [105]. However, due to their hydrophobic nature [106] and their affinity for hydrophobic substances [107], CNTs’ cylindric architectures can strongly bind to the functional groups of organic xenobiotics [108]. Additionally, upon adsorption, CNTs can also remove organic pollutants by catalyzing advanced degradative oxidation reactions (ADORs). Adsorption mechanisms consist of pore diffusion, surface diffusion and adsorption reactions [109].

Pesticides, pharmaceuticals and residual antibiotics are organic xenobiotics that can enter the environment via WW and are highly harmful to the ecosystem even at minimal concentrations. Pesticides can alter the hormonal equilibrium in plants and animals, causing detrimental outcomes for health and potential death [110]. On the other hand, pharmaceuticals and antibiotics are toxic and can disturb the endocrine system. In Appendix A, case studies concerning the application of raw and modified CNTs for the deletion of pesticides, pharmaceutical compounds and emerging organic xenobiotics have been reported. In particular, Appendix A includes experimental works on the application of CNTs to remove organic pollutants by adsorption, with the related optimized operational conditions and data of removal efficiency (RE%) and maximum adsorption capacity (MAC, mg/g).

Moreover, among all possible organic contaminants, dyes, especially azo dyes, are the most common non-biodegradable organic substances detectable in WW that are dangerous to the environment and living organisms [106,111]. In particular, the widespread modern textile production associated with inadequate WW treatments has dramatically increased the discharge of toxic and carcinogenic dyes and pigments into the environment [111].

Even if dyes are extremely toxic products of industrial manufacture, they are widely employed in the production of several goods, mainly for aesthetic purposes. As part of the water effluent from industries, dyes endowed with exceptional stability are harmfully discharged into natural water systems, and due to their partial or complete solubilization, they give an unpleasant color to water [112]. Their color is so intense that even very low concentrations of dyes in effluents are highly visible [111]. Worryingly, dyes can cause serious disorders in the environment, plants, aquatic inhabitants, humans and animals due to their toxicity, mutagenicity and carcinogenicity [111]. Dyes are resistant to natural photochemical degradation and obstruct sunlight penetration, leading to incomplete photosynthesis, thus altering the natural equilibrium of marine and freshwater ecosystems [113,114]. In particular, these events limit the essential functions of marine and freshwater ecosystems for life. Indeed, among their several pivotal operations, the components of the aquatic ecosystem filter, dilute and store water, prevent floods, maintain climate balance locally and globally, and protect biodiversity [111]. Furthermore, when in water, dyes stimulate the growth of many aquatic microorganisms that are lethal to humans and aquatic life [111]. Additionally, the byproducts of an incomplete degradation of dyes by simple photolysis, not leading to dye mineralization, are toxic, carcinogenic [115] and mutagenic [114]. Both dyes and several of the products of their photolysis are capable of depleting oxygen levels in water, with detrimental consequences for water-living organisms [116,117]. As an example, the postulated pathway of the UV/nitrate photocatalytic degradation of methylene blue (MB) up to mineralization as reported by Luo et al. [118] is shown in Figure 1.

Among all 15 products identified by the authors before mineralization to NH_4_^+^, NO_3_^−^, CO_2_ and H_2_O, the molecules shown in the first two squares are toxic to the environment like MB and can remain in water after simple photolysis. In this context, CNTs have been unequivocally demonstrated to have high efficiency in the removal of dyes from WW, both by non-degrative adsorption [114,119,120,121,122] and photocatalytic degradation up to mineralization, thus not leaving toxic intermediate residuals. Concerning absorption, issues such as possible strong π–π stacking interactions between CNTs negatively affect their removal efficiency [70,123]. Additionally, the necessity to fully remove CNTs from water after adsorption, to avoid possible health and environmental risks [124,125], further compromises their extensive utilization. To address these issues, CNT modification/functionalization by several methods, such as oxidation, magnetization, polymerization, magnetic polymerization, CVD, etc., or using natural products, has significantly enhanced CNTs’ adsorption capacity. Regardless, as further discussed in the following section, modified CNTs have demonstrated even higher efficiency in water decontamination, by the photocatalytic degradation of dyes and other organic xenobiotics, which implies advanced oxidation processes (AOPs). AOPs have many advantages over other degradative techniques, such as simple photolysis. The photocatalytic process has domain steps in the degradation pathways to mineralization, including demethylation, ring shortening, ring opening, hydroxylation and addition of NO_2_ radicals, depending on the photocatalytic conditions [118,126]. Practically, the pollutant molecules are mineralized into inorganic, simple and non-toxic molecules, such as SO_4_^2−^, NH_4_^+^, NO_3_^−^, CO_2_ and H_2_O, rather than being transformed only into other toxic byproducts, as occurs in simple noncatalytic photolysis. AOPs can occur in a wide range of pollutants, including pathogens. AOPs go beyond producing secondary pollutants, but small inorganic ions are produced as final products [127]. Appendix A collects a series of case studies concerning the removal of several dyes by adsorption, using both raw and modified CNTs. The synthetic methods used to prepare CNTs, optimized operational parameters and data of removal efficiency (RE%) or adsorption capacity (AC, mg/g) are also included in Appendix A. Other studies that merit mention include a study by Sobhanardakani et al., where HNO_3_ was employed to oxidize MWCNTs, thus improving adsorption of Janus green dye at pH = 5 in 60 min [128]. Almoisheer et al. prepared an absorbent nanocomposite made of graphene oxide (GO), SiO_2_ and SWCNTs via a hydrothermal method, and it demonstrated the capability to absorb Congo red (CR) dye from WW [129]. The modification of CNTs via surface functionalization employing a highly efficient oxidizer created –CO, –OH or –COOH groups on the external layer of CNTs, which worked as regions for the natural construction of layered double hydroxide (LDH) materials onto the surface of CNTs [130]. LDHs demonstrated outstanding efficiency in dye adsorption due to their large surface area, thermal stability, regeneration efficiency and low cost [114]. Gholami et al. synthesized -OH- and -COOH-functionalized MWCNTs, which were used to modify a polyvinylidene difluoride (PVDF) membrane, providing adsorbent materials that were successful in the treatment of dye-polluted WW [131]. Zn-Al LDH attached on MWCNT–sodium dodecyl sulfonate was synthesized and successfully experimented with by Zong et al. in the removal of CR dyes [132]. Numerous other advanced strategies and pieces of equipment have been developed to improve the adsorption efficiency of CNTs for organic dye removal, including magnetization techniques (Gabal et al. (2018)) [133], polymerization approaches (Eskandarian et al.) [134], functionalization with glycine-β-cyclodextrin (Mohammadi and Veisi) [135], magnetic separation (Duman et al. [136] and Boukhalfa et al. [137]), chemical deposition (Qureshi et al.) [138] and the use of NiO NPs and graphite structures [139,140,141,142,143,144,145,146]. All these techniques and CNT-based nanocomposites have efficiently contributed to adsorbing and eliminating organic dyes from water, thus addressing the significant environmental challenge posed by dye pollution in industrial WW. Additionally, a substantial part of research in the field was and is focused on the design of more environmentally friendly modifications of CNTs, including the use of plant-based materials. CNTs functionalized with *Saccharum munja* plant biomass were experimented with by Yadav et al. as an adaptable bio-adsorbent for the removal of MB [147]. The findings from these experiments evidenced the potency of associating plant-derived materials with CNTs to engineer eco-friendlier and more efficient bio-adsorbents for removing dye from WW.

#### 2.1.3. CNTs for Organic Xenobiotic Removal by Photocatalytic Degradation

Photocatalysis has been extensively experimented with over the years as a method for degrading organic xenobiotics, thus realizing the detoxification of water. Metal nanoparticles (MNPs) and metal oxide NPs (MONPs), mainly including ZnO, Fe_2_O_3_, CdS, and TiO_2_ NPs, are well-known photocatalysts. However, if used alone, these semiconductor materials demonstrate several issues that limit their efficiency as photocatalytic devices. TiO_2_ NPs can only be activated by UV irradiation, and therefore, they do not work efficiently under the visible irradiation of sunlight [148]. ZnO and CdS photocatalysts are prone to photo-corrosion, which translates into reduced stability and photoactivity. Also, all semiconductor NPs have a low quantum efficiency due to the fast recombination of holes (h^+^) and electrons (e^−^) [149]. CNTs have demonstrated the capability to increase the performance of MONP photocatalysts. In fact, when CNTs are inserted as ingredients in a reaction mixture to prepare nanocomposite photocatalysts, they strongly augment the MONPs’ catalytic efficiency due to their excellent optical, electrical and mechanical capabilities [150,151]. Depending on their diameter and chirality, CNTs can be semiconducting or metallic [150].

##### CNT-Assisted Photocatalytic Degradation of Organic Pollutants: The Proposed Mechanism

When light activates an MONP photocatalyst, electrons (e^−^) shift from the valence band (VB) to the conduction band (CB), leaving cationic charges (h^+^) in the VB. In the absence of CNTs, some of the cationic and anionic charges combine immediately, thus extinguishing each other and conferring MONPs a poor quantum efficiency [149]. This phenomenon prevents uncombined e^−^ and h^+^ from initiating the photocatalytic oxidative degradation of organic contaminants, leading to a low degradation efficiency. On the contrary, when MONPs are merged with CNTs, forming a nanocomposite photocatalyst, strong interactions between CNTs and MONPs occur, resulting in tight connections, with the formation of a barrier junction (Schottky barrier); these connections impede the early recombination of h^+^ with e^−^ and promote the transit of h^+^ to the VB (Figure 4).

CNTs act as e^−^ acceptors, which seize e^−^ deriving from the semiconductor MONPs, which instead act as e^−^ donors when exposed to light. Uncombined e^−^ are free to react with oxygen molecules, thus generating highly reactive superoxide radical ions, which carry out the oxidative degradation of pollutants. Additionally, the uncombined h^+^ can further degrade the pollutants by oxidizing hydroxyl groups (OH) to form hydroxyl radicals (•OH) [14,152] (Figure 4) [85]. Moreover, CNTs can inhibit the aggregation of MONP crystals, thus improving their photocatalytic activity and enhancing the adsorption of contaminants on MONP photocatalysts, due to their high-quality active sites and large surface area [14,152]. CNTs also have a stronger affinity for e^−^ in the CB on MONPs and speed up their relocation from the CB to the CNT surface due to their lower Fermi level [152]. CNTs promote the adsorption of more OH groups on the surface of MONPs, which are oxidized by the h^+^ to generate •OH [152]. In particular, it has been reported that XPS results showed that there are more hydroxyl groups provided by chemisorbed water on the surface of a TiO_2_/CNT composite due to the higher adsorption ability offered by CNTs, deriving from their large surface area. This phenomenon is advantageous for increasing the photocatalytic activity of MONPs. The more hydroxyl groups on the surface of a photocatalyst, the more hydroxyl radicals will be produced by the oxidation of h^+^ [153]. By merging CdS with CNTs, photo-corrosion can be limited due to the enhanced adsorption capability of CNTs against reductants in the solution [149,152].

Nanogold particles (Au NPs) are visible region photosensitizers, but their photocatalytic efficiency is limited because of a fast, early recombination of h^+^ and e^−^. The combination of Au NPs with CNTs remarkably enhanced the photocatalytic functions of Au NPs, providing efficient visible light photocatalyst nanocomposites [152,154].

Since SWCNTs have more individual contacts than MWCNTs, they can improve the photocatalytic activity of TiO_2_ more than MWCNTs. However, since SWCNTs have a high resistance, it is necessary to lower the interface charge transfer resistance of SWCNT/TiO_2_ [152]. Appendix A collects a series of case studies on the application of CNTs/MONPs nanocomposites for the photocatalytic oxidative degradation of different organic contaminants in WW.

#### 2.1.4. Application of CNTs in Removing Pathogens and Cyanobacteria Toxins from Water

In addition to the previously reported xenobiotics, which constitute the most part of water pollutants, microorganisms (e.g., bacteria, viruses, protozoa, etc.) and biological toxins (e.g., cyanobacterial toxins) may also contaminate water. Cyanobacteria are toxin-producing bacteria that can contaminate water. During water purification treatments, they can undergo lysis, thus releasing their toxins into the water and producing further contamination. Microbial contaminants encompass human pathogens, including *Salmonella* spp., *Shigella* spp., *Escherichia coli*, *Vibrio cholera*, *rotavirus, Cryptosporidium parvum*, *Giardia cysts*, etc., deriving from inflowing water where sewage not opportunely treated at the source is discharged [155,156,157,158] Additionally, free-living microbes naturally resident in the source, such as *Legionella pneumophila*, *Microcystis aeruginosa*, *Pseudomonas aeruginosa*, etc. [159,160], nuisance organisms, including iron- and sulfur-reducing bacteria [158], and microorganisms causing taste and odor issues, such as *Vannella*, *Saccamoeba* and *Ripidomyxa*, can pollute water. There is also an exhaustive list of biothreat pathogens, classified as potential contaminants, that are currently absent in water but may contaminate it upon an unfortunate incident of a bioterrorism attack [158,161]. Cyanobacterial toxins are produced by species of toxic cyanobacteria, which produce harmful algal blooms (HABs). These bacteria live mostly in the non-turbulent fresh water of lakes and rivers worldwide, except for Antarctica [158]. Cyanobacterial cells have special gas-filled vesicles inside their vacuoles, where a constant filling and emptying mechanism takes place, gifting the bacteria with a buoyancy capacity, which allows millions of cells to migrate to the water surface where they form a layer of crust [158]. When they are transported into treatment systems, the cells can be lysed, and the intracellular toxins, such as microcystin derivatives (MCs), can be released into the water, conferring an unpleasant odor and taste. The intrinsic antibacterial nature of pristine CNTs [2] inhibits the growth of pathogens on their surface, thus promoting the self-cleaning efficiency of CNT-based adsorption filters used to decontaminate drinking water from microorganisms. In comparison to filters made of granular activated carbon (GAC), CNT-based filters deactivate microorganisms rapidly on contact after adsorption. On the contrary, GAC-based filters are not capable of killing pathogens, which are free to grow and create biofilms on their surface [162], with the possible release of microbes in the effluent water [163]. Unmodified and functionalized CNTs have demonstrated antimicrobial characteristics over a wide range of microorganisms, including bacteria, such as *Micrococcus lysodeikticus* [164], *Streptococcus mutans* [165], *E. coli* [166,167,168] and *Salmonella* [169], bacterial endospores [170], protozoa species, such as *Tetrahymena pyriformis* [171], and viruses, such as bacteriophage MS2 [172]. Additionally, the residual lysed cells can be removed from the filters with backwashing. Collectively, the antimicrobial characteristics of CNTs indirectly improve the microbial sorption efficiency, providing simultaneous capture and deactivation of pathogens. As recently reported, CNTs are antimicrobial due to their fibrous shape [2]. Their thin fibers interact with the bacterial cell surface, damaging its cytoplasmic membrane and/or cell wall, disrupting the intracellular metabolic pathways and inducing ROS overproduction, causing oxidative stress (OS), cell rupture and cell content release [2]. The size and length of CNTs, their dispersibility, the content of amorphous carbon and the number of layers differentiating single- or multi-walled CNTs (SWCNTs or MWCNTs) influence CNTs’ cytotoxic properties to microbes more than the presence of metal impurities deriving from the catalysts necessary for their preparation [166,167,168,169,173,174]. Based on literature data, the toxicity to pathogens is high in the case of uncapped, de-bundled, short-length and highly dispersed MWCNTs. In general, SWCNTs demonstrate higher antimicrobial effects than MWCNTs due to their short length, which allows them to easily penetrate through the cell membrane and impair the pathogen’s vital processes [168]. Furthermore, when CNTs are highly dispersed, a larger cell contact can occur, thus increasing the rate of cell lysis. Additionally, CNT functionalization leads to an increase in their water solubility and therefore their dispersibility, thus making hydrophilic CNTs capable of providing better surface contact with biological adsorbates, including microbes, than hydrophobic CNTs, thus enhancing their toxic effects towards pathogens [175]. Moreover, the dispersibility of CNTs is essential for allowing their use in the fabrication of CNT composite membranes. On the other hand, Akasaka and Watari demonstrated that the efficiency of the capture and precipitation of bacteria by CNTs mainly depends on the balance between adequate dispersibility and aggregation activity of CNTs, which in turn is a function of their diameter [165]. Specifically, the authors experimented that the precipitation of *S. mutans* isolates on 30 nm semi-dispersible MWCNTs was greater than that caused by completely dispersed SWCNTs and weakly dispersible 200 nm MWCNTs [165].

##### Adsorption of Microorganisms on CNTs

Unlike microporous materials commonly used for removing pollutants from drinking water, such as powdered activated carbon (PAC) and graphene activated carbon (GAC), which are poorly efficient in concentrating microbes, CNTs showed exceptional and unprecedented high bacterial adsorption capacities. Additionally, CNTs demonstrated selectivity for bacteria, with almost instantaneous adsorption kinetics, making them ideal materials for developing pathogen sensors, which require rapid bacterial concentration. The ability of pristine SWCNTs to adsorb spores of *Bacillus subtilis* was compared to that of PAC and NanoCeram™ by Upadhyayula et al., confirming the higher affinity (27–37 times greater) of SWCNTs for these isolates [176]. As reported, the high adsorption efficiency of SWCNTs was mainly due to their fibrous size and major accessibility of external surface area [177]. In summary, fibrous adsorbents such as CNTs have better outcomes in the sorption of microbes than powdered adsorbents [178]. Selective bacterial adsorption by CNTs was demonstrated by Deng et al. [179] using pure and mixed bacterial cultures of *Staphylococcus aureus* and *E. coli* strains and different concentrations of unmodified SWCNTs [179]. The results evidenced that the sorption efficiency of *S. aureus* was 100 times greater than that of *E. coli*, that mixed culture adsorption data correlated well with the pure culture Freundlich isotherm equation and that CNTs selectively concentrated one species of bacteria (*S. aureus*) over the other [179]. The rapid kinetic rates for the absorption of *B. subtilis*, *S. aureus* and *E. coli* by SWCNTs from water at an initial bacteria concentration higher than 10^7^ CFU/mL were studied by several authors and reported in 85 articles before the year 2010 [158]. The results unequivocally demonstrated that >95% of bacteria were concentrated by SWCNTs in a time range of 5–30 min. Rapid kinetics and high bacterial sorption capacities established that CNT-based point-of-use (POU) filters can be deployed for unlimited uses in the removal of microorganisms from raw waters. CNTs are ideal materials for bacteria removal due to their strong antimicrobial activity. However, relevant research in water treatment and environmental remediation is limited. Brady-Estévez et al. demonstrated that an SWCNT-enriched polyvinylidene difluoride (PVDF) membrane was capable of retaining and killing *E. coli* and the MS2 bacteriophage virus in water [180,181]. The antimicrobial action of SWCNTs toward *E. coli, P. aeruginosa, B. subtilis* and *S. epidermidis* was successfully evaluated by Khan et al. both in river water and WW effluent [182]. The synergistic, stronger sporicidal effect of SWCNTs combined with oxidizing antimicrobial chemicals such as H_2_O_2_ and NaOCl on *B. anthracis* was shown by Lilly et al. [183]. Nanocomposites encompassing SWCNTs or MWCNTs and metal particles such as silver, ZnO, Ti_2_O and ferric oxide NPs exhibited excellent synergistic antimicrobial efficiency in water treatments, leading to improved water decontamination from viruses, *E. coli, P. aeruginosa* and *S. aureus* and disinfection compared to NPs and CNTs alone [184,185,186,187,188]. Antimicrobial polymers coated on both SWCNTs and MWCNTs provided membrane nanocomposites with improved efficiency in water disinfection, depending on their different molecular size, chemical composition and physicochemical properties of functional groups [189,190]. Ahmed et al. developed polyvinyl-*N*-carbazole (PVK)–SWCNT nanocomposites with different SWCNT contents that showed remarkably antimicrobial effects against pathogens [191,192]. SWCNTs attached to the surface of thin film allowed up to 60% inactivation of *E. coli* bacteria within 1 h of contact time [193], while MWCNTs merged with aromatic polyamide or polyether sulfone gave nanocomposite membranes for fouling control and microbial removal in water treatment [194,195].

##### Adsorption of Cyanobacterial Toxins on CNTs

It has been reported that conventional water treatments and disinfection techniques, such as coagulation/flocculation processes, as well as the use of disinfectants, such as chlorine and other strong oxidizers, such as potassium permanganate, ozone and hydrogen peroxide, have limited success in cyanobacterial toxin (CT) removal from water systems [196,197]. On the contrary, adsorption methods, using natural materials including clay minerals, sepiolite, kaolinite, talc and activated carbons (ACs), were more efficient in decontaminating water from CTs [197,198,199,200,201,202,203]. ACs, in the forms of both granular activated carbon (GAC), coconut-based powdered activated carbon (CPAC) and wood-based activated carbon (WPAC), are the most efficient materials [197,202]. GAC reduced CT levels by 95%, whereas WPAC by >95%. Moreover, GAC demonstrated shorter service lifetimes than PAC [196]. The higher adsorption efficiency of PAC depends mainly on the presence of 2–50 nm mesopores, which improve the intraparticle diffusion of CTs [197,202,203]. In addition, the surface chemistry, the pH of the solution, and the presence of additional organic matter pollutants, acting as competing items, can affect the outcomes of the adsorption process. The surface of AC is mostly dominated by negatively charged anionic functional groups (carboxylic and hydroxyl) [197]. Low pH values up to 2.5 favor the adsorption procedure, while competing elements, such as algal and extracellular fractions, usually decrease the efficiency of AC, due to their higher polarity with respect to CTs [197]. Additionally, if water is chlorinated, AC-based materials’ capability to remove CTs is further reduced because chlorination converts certain organic xenobiotic compounds to hydrophilic organic compounds, which are preferentially dislodged from water [197]. Considering these results, many researchers were stimulated to test CNTs as new adsorbent systems to reduce CT levels in water, and they observed that CNTs are superior to AC and are even 4 times more efficient than natural clays [204]. Some studies reported that the higher adsorption capacities of CNTs depend on their high mesoporous volume [202,204], high specific surface area [205], symmetrical structure [206], presence of surface defects [206] and external diameter [202]. Ye et al. showed that the adsorption of microcystins and uremic CTs on CNTs was exceptionally high [206]. It occurred first via the diffusion of CTs from the bulk into the mesoporous areas of CNTs and then through their robust adhesion to the CNT walls’ pores, thus passing from the adsorbing phase to the adsorbed one [206]. The same authors demonstrated that the adsorption of CTs on CNTs was best modeled by the Langmuir isotherm and follows a monolayer pattern [202]. AC, possessing a microporous structure, demonstrated very low kinetics, allowing an equilibrium to be reached in more than an hour, due to the high resistance to the inner diffusion encountered by adsorbates. Conversely, CNTs having highly mesoporous structures showed very fast kinetics. This was due to the minimal resistance that adsorbates had to face for diffusion, thus allowing an equilibrium to be reached in less than 10 min [202,206]. Additionally, CNTs with outside diameters in the range of 2–10 nm showed higher adsorption efficiency towards CTs having dimensions ≤ 1.9 nm, while desorption processes from tubes with smaller diameters were less efficient than those from tubes with larger outside diameters [202]. The possible presence of organic material as co-pollutants (OMPs) can be helpful in removing CTs [200]. In fact, when OMPs are present, they can support the attachment, formation and long-term duration of bacterial biofilms on adsorbents, thus helping the biodegradation process of CTs by the bacteria, which can utilize CTs as support for their metabolic functions [200]. However, pristine CNTs are normally not suggested for this bioremediation strategy, because their cytotoxic effects inhibit bacterial growth. Conversely, Yan et al. demonstrated that opportunely synthesized biocompatible CNTs can allow the growth of bacteria, thus permitting the bioremediation of CTs [204]. Specifically, it was reported that *Ralstonia solanacearum* grown on biocompatible CNTs demonstrated a bioremediation capacity of CTs from water 20% higher than that of *R. solanacearum* alone [204].

#### 2.1.5. Water Treatments Using CNT-Based Filtering Membranes

The several advantages that could derive from enriching traditional filtering membranes with CNTs are schematized in Figure 5.

Due to the nonpareil features of CNTs, filtering membranes enriched with CNTs have demonstrated better selectivity and permeability and could substitute and improve the performance of traditional membranes commonly used for water decontamination [150,151,207]. CNTs help the polar water molecules to flow quickly, due to their hydrophobicity, thus allowing experts in the field to produce novel CNT-improved filtering membranes requiring little or even no energy for filtration. Currently, CNT-based membranes comprise composite, buckypaper and vertically aligned carbon nanotube (VACNT) membranes [208]. Moreover, by proper functionalization, the hydrophobicity of CNTs can be further enhanced, allowing the faster filtration of more water, thus increasing the overall water permeability of the conventional hydrophobic membranes. Moreover, CNT-enriched membranes possess enhanced antifouling capabilities [207,209].

Liu et al. reported that the addition of CNTs to conventional membranes minimized their fouling drawbacks [210], while further chemical modifications of CNTs, altering their surface by inserting functional groups, improved supplementary antifouling capabilities, selectivity–permeability trade-offs, surface roughness and porosity variation [207,209,210].

By increasing the number of pores per unit area in the CNT structure, a higher pore density can be achieved, thus improving the permeability of CNT-improved membranes. In this regard, VACNTs were densified utilizing mechanical densification or capillary-force procedures, thus obtaining a higher number of pores in a VACNT-based membrane. However, in addition to pore density, other factors are critical for high membrane permeability, such as the number of CNT layers and CNT size [207,209].

According to many authors, CNTs are capable of enhancing the separative performance of conventional polymer-based membranes, due to their high flexibility, the very low friction coefficient of their internal layers, their high aspect ratio, the presence of many nanochannels and their efficient interaction with aromatic pollutants [207,209,211].

Additionally, the insertion of CNTs in filtering membranes led to enhanced mechanical strength, selectivity, faster flux, super-hydrophobicity, improved wetting resistance, stability over time and higher permeability [151,207,209]. Appendix A collects a series of laboratory studies that employed different filtering membranes modified using diverse types of CNTs to treat WW, highlighting the preparation methods, filtration performance and contaminant removal efficiency. As mentioned for other characteristics of filtering membranes enriched with CNTs, the introduction of proper functional groups on the CNT surface further improved membrane wetting behavior [207,209]. Unfortunately, such modifications could damage the CNT structure, resulting in low efficiency and a low flux rate [207,209]. Among other drawbacks, the hydrophobicity of CNTs determines their preferential aggregation, which complicates the fabrication of CNT-based composite membranes [207,209].

Finally, until now, CNT-based membranes have been employed to filter modest quantities of WW in lab-scale batch experiments. Further work, including experiments with real environmental WW, in place of appositely created laboratory water in batch experiments, is necessary to improve the CNT-based filtration technology and reach the large-scale, extensive application of CNT-based membranes [207,209].

Among the different types of CNT-based membranes developed, vertically aligned CNT (VACNT) and buckypaper (BP) membranes are the most produced. VANCTs provide membranes typified by a regular alignment of CNTs perpendicular to the membrane surface [212], while BPs are substantially characterized by a structure containing large pores in which CNTs are casually arranged, forming a tangled network. The random arrangement of CNTs in BPs weakens their mechanical robustness and lowers their electrical performance. Membranes based on VANCTs possess higher mechanical strength and excellent thermal and electrical conductivity, present controlled porosity and provide high flux [213]. Differently, BP membranes are characterized by a very cohesive and strong architecture, possess a large surface area and demonstrate high porosity (60–70% of the total volume). These characteristics are possible because, despite the CNTs in BPs being assembled only by π-π and energetically low Van der Waals (VDW) interactions, their number of VDW interactions per volume unit is tremendously high. The mechanical strength of a BP membrane can be improved by the alignment of CNTs and by CNT functionalization. Oxidized CNTs showed a tensile strength 6.3 times higher than that observed for unmodified BPs. Also, other factors, such as diameter, length and the utilization of SWCNTs rather than MWCNTs, as well as the adopted procedures for fabricating the membranes, can affect the properties of both types of CNT membranes [214,215]. The major unsolved issues of VACNT membranes, hampering their extensive practical application, regard the high cost of production and the need to use strong chemicals for production [209,215]. On the contrary, the available procedures used to fabricate BPs are smoother and quicker, easily scalable and cost-effective [215,216,217]. The methods for preparing BPs include mainly wet (vacuum filtration) and dry methods, with the former being the most adopted technique. Being substantially different, BP and VACNT membranes are usually applied for different purposes. Due to their aligned structure, which is responsible for high permeability and high rejection rates, VACNT membranes are preferentially employed in separation processes. However, due to the remaining unsolved issues related to the complex and high-cost production of VACNTs, BP membranes are mostly applied.

In fact, due to their easy and low-cost fabrication, but also to their excellent properties, including large surface area, light weight, high flux and chemical tunability, BP membranes represent ideal tools for treating wastewater. BPs are remarkably hydrophobic, which makes them suitable for organic large-molecule pollutant removal (Table 4) and desalination [218,219,220], and they have demonstrated a pronounced antibacterial activity (Table 4).

Table 5 below compares several characteristics of membranes based on original CNTs, VANCTs and BPs.

##### CNT-Based Membranes for Water Desalination

The possible application of CNT-improved filtering membranes for water desalination has been extensively examined due to the high surface area/volume ratio and chemical and mechanical robustness that CNTs can confer membranes for desalination [150,210]. The reported findings concerning the use of VACNT-based and composited CNT membranes in water desalination have evidenced enhanced mechanical features, high selectivity, advanced salt removal efficiency, and improved water flux [57,236].

The process of transport and salt separation assumed for CNT-based filtering membranes is schematized in Figure 6.

The desalination technique utilized to separate salt from water necessitates a phase shift from liquid to vapor. In this regard, due to their high hydrophobic nature, CNTs favor vapor perviousness, while preventing liquid from entering the membrane pores. Additionally, since CNTs possess fast adsorption and desorption capacities, water vapor molecules can undergo surface diffusion (route (1) in Figure 6). During this, salt molecules jump from one site to another via interaction with the membrane surface. Moreover, CNTs provide alternative routes for the separation of water vapor molecules from salt ones, both via transport by diffusion along their smooth surface (route (3) in Figure 6) and via fast transport directly through their inner channels (route (2) in Figure 6).

Compared to the typically used desalination membranes, CNT-improved ones allow easy water desalination at lower temperatures and possess higher resistance to strong saline concentrations up to saltwater levels. Since the absorption of ions and water by CNTs is strongly affected by the electrostatic charge distribution, by altering the alternating charge design of the CNT channel, it is possible to encourage water intake and inhibit ion absorption [57,236,237]. Appendix A summarizes some relevant studies on the application of CNT-enriched filtering membranes for water desalination. The preparation methods (PMs), the optimized operational parameters (OOPs), the main contribution of CNTs and data of removal efficiency (RE%) are included in Appendix A.

Some authors have reported that the higher water desalination performance of CNT-improved membranes is mainly due to the huge surface area, high surface-to-volume ratio, chemical robustness and high mechanical strength possessed by CNTs. Moreover, functionalized CNT-based membranes proved enhanced salt rejection capability due to chemical interactions, the Donnan effect and electrostatic interaction between salt ions and CNTs [151,236]. Wang et al. demonstrated the several advantages of using CNTs in desalination processes, including increased water flux, antifouling capacity, excellent selectivity, good sieving capabilities and excellent mechanical strength [236]. However, the same Wang et al. highlighted also some CNT-associated issues that limit the extensive application of CNTs in water desalination membranes, including the high commercial cost, which is particularly unsustainable for SWCNTs [236].

### 2.2. CNTs for Detection and Remediation of Gaseous Pollutants

Air quality is significantly compromised by the presence of hazardous gaseous pollutants, which are mainly byproducts of human-induced anthropogenic activities [84] and have irreversible chronic toxic effects on human health. This is the case for several oxides, including those of carbon (CO and CO_2_), nitrogen (NO_x_) and sulfur (SO_x_), as well as ozone (O_3_), methane (CH_4_), volatile organic compounds (VOCs), particulate matter, etc. Pollutants, such as greenhouse gases, contribute to global climate change and global warming. In this scenario, miniaturized gas sensors are necessary to monitor air quality. Although endowed with high sensitivity, conventional solid-state gas sensors exhibit problems, such as high working temperatures and poor long-term stability [238]. CNTs were used in the preparation of miniaturized gas sensors, demonstrating significant advantages, including rapid response, enhanced sensitivity, significant gas adsorption capability and a large active surface area [239,240]. Additionally, CNT-based sensors displayed low power consumption and operated well at lower temperatures. Excellent interactions with organic compounds were possible via non-covalent forces, including electrostatic forces, as well as VDW, H-bonding, π-π and hydrophobic interactions [241]. The selectivity and stability of CNTs can be further enhanced by incorporating functional groups (OH, COOH, CO) [242]. These distinctive features make CNTs an attractive option as an active material for gas sensing applications in the configurations of chemo-resistors and field-effect transistor chemical sensors [94,243]. Kong et al. created a CNT-based gas sensor with electrical conductivity that displayed sensitivity and prompt responses to NO_2_ and NH_3_ [244]. Rigoni et al. reported that SWCNT-based chemo-resistors presented excellent sensitivity to NO_2_, with a limit of detection as low as 20 ppb [245]. Sayago et al. investigated various DWCNTs and MWCNTs materials as resistive NO_2_ gas sensors, thus developing innovative sensor films that sensed NO_2_ at concentrations as low as 0.1 ppm [246]. Kumar et al. engineered a gas sensing mechanism using SWCNT-based adsorbents, amplifying the chemisorption of NO_2_ molecules [247].

By using MWCNTs, a reversible chemo-resistor that detected methane at ppm levels with high sensitivity was obtained by Kathirvelan and Vijayaraghavan [248], while Star et al. used SWCNTs functionalized with metal nanoparticles (NPs) for the same purpose [249]. Jung et al. experimented with the preparation of a sensor based on MWCNTs, capable of sensing H_2_ at low concentrations [250]. The authors demonstrated that thermal treatment of the exhaust sensor at 1000 °C promoted H_2_ desorption, thus realizing the recovery of the sensor for reuse. In a study by Kong et al., a sensor for detecting H_2_ at ppm concentrations was developed by the electron beam evaporation technique for palladium (Pd) deposition onto SWCNT surfaces [244].

By CNT functionalization with sulfonated and carboxylic acid moieties, sensors capable of detecting carbon monoxide (CO) at a 1 ppm concentration were manufactured, thus meeting the urgent need for devices detecting CO at extremely low levels, due to CO’s high toxicity [251,252].

Additionally, CNTs coated with gold (Au) NPs were capable of detecting carbon dioxide (CO_2_) with high sensitivity, as well as other volatile organic compounds, including ethanol (EtOH) and isopropyl alcohol (iPrOH), as also reported by Dilonardo et al. [253]. In this case, the authors demonstrated that their Au-coated CNTs had high sensitivity to NO_2_, NH_3_ and H_2_S at temperatures in the range of 100–200 °C.

Starting from metal alkoxides and carboxylic acids as precursors, Willinger et al. used a novel atomic layer deposition (ALD) method for uniformly coating CNTs with vanadium titanium, and hafnium oxide thin films, obtaining large-surface hybrids. The electrical and sensing properties of resistive gas sensors based on hybrid vanadium oxide-coated CNTs (V_2_O_4_–CNTs) were reported, and the effect of thermal treatment on the gas sensing properties was studied [254].

By using a 10 min CF_4_ plasma treatment, Zhang et al. prepared MWCNTs with superior consistency and sensitivity, as well as with enhanced gas sensitivity towards H_2_S and SO_2_, compared to other types of CNTs [255].

Despite being time-consuming and possessing low thermal cyclability and a limited lifespan, plasma, electrochemical, mechano-chemical or chemical treatments were used to modify the chemical composition of the surface of CNTs’, thus enhancing their gas sensitivity [256,257,258,259]. Specifically, CNTs’ sensitivity to benzene, NO_2_, NH_3_, H_2_, H_2_S and CH_4_ was improved by metal nanocluster deposition on their surface [249,260,261,262,263,264,265], while that to CO was enhanced by oxygen plasma treatment [266].

On the contrary, a doping technique allowing the introduction of dopants during the growth of CNTs, thus avoiding boring post-synthesis steps, was an efficient strategy for tuning the sensitivity of CNTs to gases [267,268,269]. CNTs doped with boron (B-CNTs) and nitrogen atoms (N-CNTs) demonstrated modified physical and chemical properties, altered solubilities and increased surface reactivity [270,271].

Air filtration is used to separate solids from gases, commonly dust from air, and the mechanisms of particle filtration include gravitational, impaction, interception, diffusion and electrostatic [272]. Air filtration techniques can be divided into induct devices (filters) and portable air cleaners (PACs). Filters work for the entire house but function only during the operation of air-handling systems, where they are installed [272]. PACs can operate and can be positioned in a room, with flexibility to target problem areas, thus receiving increasing attention in recent years [272]. In this context, CNT-based air filters attracted the attention of researchers worldwide because of their capacity to efficiently remove gas pollutants, microplastic particulates (MPs), particulate matter (PM) and polioviruses from air, thus improving its quality. CNT-based filters’ efficiency is due to their high adsorption capability and large specific surface area [61]. Concerning their filtering modalities, several reports showed that the filtration via hydrophobic MWCNTs occurs through various mechanisms such as interception (particles of about 0.05–1 µm), inertial impaction (particles of about 0.1–1 µm), diffusion (particles of about 0.01–10 µm) and gravity (1–10 µm) [273].

Effective CO_2_ removal was achieved by embedding amines into CNTs. In this regard, Osler et al. showed that the capability of SWCNTs to sorb CO_2_ was higher than that of MWCNTs [274]. The high number of amine residues on the CNT surface enabled the efficient adsorption of CO_2_ even at lower temperatures (20–100 °C) [275]. Cost-effective and highly sensitive synthesized nickel-based MWCNTs were efficient in removing NO_2_ [276]. Santucci et al. studied, both theoretically and experimentally, the alteration of CNT thin films’ conductivity for the effective adsorption of NO_2_ and CO [277]. The deposition of CNTs on quartz filters resulted in nanomaterials capable of removing volatile organic compounds (VOCs) via π-π interactions [278,279]. Moreover, SWCNTs/NaClO were used to remove iPrOH by physical adsorption, supported by Van der Waals forces, and chemical adsorption onto the functional groups on SWCNTs [280]. MWCNTs for SO_2_ adsorption were experimented with by several authors [281,282], including Iraji et al., who, according to Ogunsola et al., modified MWCNTs through 3-[2-(2-aminoethylamino) ethyl amino]propyl tri-methoxy-silane functionalization [69], and Zhang et al., who employed a sol–gel method to synthesize three different adsorbents (MWCNT/TiO_2_ + Cu, MWCNT/TiO_2_ + Cr, MWCNT/TiO_2_ + Zn) [281]. Yang et al. proposed a low-cost and environmentally friendly procedure for synthesizing MWCNTs from plastics and red mud [282,283].

### 2.3. Regeneration of Exhausted CNTs

Appendix A summarizes the most relevant methods experimented with for the regeneration of CNTs, including the number of cycles and the main result after the last desorption cycle. The possibility to regenerate exhausted CNT-based absorbents and filtering membranes for reuse is of paramount importance to meet the sustainability and cost-effectiveness of environmental remediation. In this regard, efficient regeneration is mandatory to ensure high adsorption capacity upon several utilization and regeneration cycles [284]. Numerous studies have demonstrated that the desorption of contaminants from the surface of utilized CNTs represents an efficient method for their regeneration [285]. To this end, several regeneration techniques have been experimented with, including chemical extraction, which consists in using specific chemicals to detach pollutants from CNT surfaces [286,287]. Unfortunately, this method can damage the CNT structure, generate waste solvents [288,289,290] and have a negative environmental impact [291]. Thermal treatments based on high temperatures for desorbing adsorbates from CNTs have been proposed [291,292]. The high energy consumption and the possible induction of oxidative reactions, leading to a carbon loss of approximately 5–10%, are the main drawbacks of this technique. Accurate control and optimization can render this method a practical regeneration technique [287,290,293]. Bio-regeneration methods utilizing biological agents or processes to remove contaminants from CNTs are environmentally friendly but have limited efficiency and inadequate large-scale applicability [294]. An attractive method, due to its environmental friendliness and cost-effectiveness, involves the use of Fenton and electro-Fenton regeneration, based on the generation of hydroxyl radicals (•OH), which effectively mineralize organic compounds adsorbed on CNTs [289,295,296,297,298]. Regeneration using ultrasonic waves, which disrupt the adsorbate–CNT interactions, causing adsorbate desorption, is an energy-efficient and environmentally friendly strategy [287,299]. Also, microwave irradiation experimented with for CNT regeneration has demonstrated the possibility of a fast and efficient desorption process [300,301]. Each of the above-reported techniques presents a series of advantages and drawbacks. The choice of the most suitable method depends mainly on the type of contaminants, the desired degree of regeneration, energy efficiency and environmental considerations. The regeneration of spent CNTs is influenced by several factors, including the pH of desorption solution, the type of absorbed contaminant, the concentration and type of desorption solution [295,302,303], temperature, which can affect desorption kinetics, desorption time, the number of adsorption–desorption cycles [304,305] and the type of contaminant–CNT interaction. Some authors reported that the desorption values of some metals were close to 100% at pH = 2, while they decreased with an increase in pH, becoming very low at pH > 5 [306]. On the contrary, anionic dyes desorbed well at high pH, while cationic dyes desorbed well at low pH values [289]. Sui et al. studied the desorption of anionic MO and cationic MB from MWCNTs, finding a maximum desorption of MO (80.2%) at pH = 13.0 and MB (79.7%) at pH = 2 [307]. Similar results were also observed for the desorption of cationic dyes, such as MB and brilliant cresol blue, from MWCNTs, where the optimum desorption was observed at pH = 2 for all dyes [308].

## 3. The Other Side of the Coin: The Paradoxical Toxicity of CNTs Towards the Environment and Living Beings

At the time of their invention in 1991 [1], CNTs were produced only at laboratory levels, mainly due to the very high production costs of the few synthetic processes available at that time [309]. For years, the use of CNTs and the contact with such nanomaterials was occasional, and the possible negative effects of CNTs on human, animal and environmental health were not hypothesized or investigated [310]. Advanced synthetic methods, such as CVD, allowed CNT production on a larger scale, forcing researchers to conduct progressive studies about the impact of CNTs on the environment and living creatures. Toxicological studies for investigating CNTs’ possible hazardous effects on health were carried out starting from the first years of the XXI century, mainly stimulated by the similarity of CNTs’ shape with that of asbestos fibers, which are recognized as a material triggering inflammation and causing lung cancer [311]. Several studies have been carried out, using in vitro and in vivo models to assess CNTs’ possible toxicity. Rodents were exposed acutely or chronically to CNTs administered by different routes. Very recently, we have reported an extensive and updated overview concerning the toxicity, cytotoxicity, genotoxicity, pulmonary toxicity and toxicity to cardiovascular and reproductive apparatus associated with exposure to different types of CNTs. Exposure can occur both in daily normal life and in a work setting for CNT manufacturers [2]. From our survey, it was evidenced that generally, the dangerous effects of CNTs are mainly associated with their shape, concentration, structural defects and the presence of impurities [2]. However, the reported case studies concerning the possible toxicity of CNTs to humans and the environment are conflicting. Considering these findings, the greatest challenge of academic and industrial researchers is the development of automated systems for synthesizing CNTs, gifted with uniform and predictable properties, thus making it possible to reduce their still not sufficiently clear toxicity. Fortunately, nowadays, this goal is extraordinarily close to being reached. Very recently, an artificial intelligence (AI)-driven platform (CARGO) was proposed by Li et al. [312]. CARCO allowed the safe achievement of a new titanium–platinum bimetallic catalyst for the synthesis of high-density, horizontally aligned carbon nanotube (HACNT) arrays. An unprecedented 56% precision in synthesizing predetermined densities of HACNT arrays [312] was reached. Regardless, it would be suggestable to consider CNTs as a new chemical nanomaterial, rather than as an allotrope of inert carbon [313]. Here, thinking it redundant and boring to rediscuss all the case studies previously reported, we only provide tables reporting the earliest and latest in vitro and in vivo investigations carried out to assess the possible dangerous effects of CNTs on the environment and living beings.

### 3.1. In Vitro Studies

Table 6 collects the early in vitro experiments carried out in the years 2003–2011 to assess the cytotoxicity of different types of CNTs using several cell lines.

### 3.2. In Vivo Studies: Pulmonary Toxicity

The early in vivo experiments carried out to assess the possible pulmonary toxicity of CTNs are summarized in Table 7.

Acute and chronic pulmonary inflammation were found in rodents after exposure to CNTs [328,329,333,334]. Based on these observations, researchers assumed that the toxic effects of CNTs could also compromise other tissues. The cardiovascular toxicity of CNTs to rodents was then investigated upon administration of CNTs in different ways. Regardless of the administration mode, activation of the blood cells triggered by the inflammatory markers released was observed. This event caused dangerous cardiovascular outcomes and provoked injury to the mitochondrial DNA of the aorta. Atherosclerotic plaques on the surface of the aorta and increased atherosclerotic lesions in the brachiocephalic arteries were observed [320,335,340]. Additionally, the toxic effects of CNTs on the reproductive and developmental system were investigated by other authors. High rates of resorption processes, progressive malformations in the surviving infant rodents, ROS overproduction in the placentas of exposed animals, tissue damage and testicle alterations were found [341,342,343]. A collection of recent reports on the in vitro and in vivo toxicological aspects of CNTs is available in Table 8, while Table 9 provides reports on CNTs’ injuriousness to several organs.

## 4. How Can the Toxicity of CNTs Be Moderated?

Upon establishment that CNTs can accumulate in the environment, thus having noxious outcomes, several authors have studied strategies for preventing the activation of the toxic systems in the tissues of organisms in contact with them [369,370,371,372,373,374]. The developed approaches are mainly based on CNT surface modifications and functionalization, with PEG, C1q recombinant globular proteins, biocompatible ingredients or molecules, which can improve their solubility and dispersity in biological fluids. Additionally, molecules capable of reducing CNTs’ capacity to induce oxidative stress (OS), such as curcumin or quercetin [375], thus preventing the oxidative damage, inflammatory and immunotoxic effects of MWCNTs, were also exploited [376,377,378]. Functionalization with -COOH and -OH groups to influence cellular uptake and interactions was carried out [379,380]. Moreover, notable advances in the purification of CNTs have been made to achieve less toxic, highly pure CNTs without defects, not containing residual metals or catalysts [381]. Additionally, more biodegradable CNTs have been developed over time to reduce their persistence in the blood and reduce tissue toxicity [382]. Regardless, limiting the toxicity of CNTs continues to be a challenge for experts in the sector. Concerning the investigations performed to evaluate the possible toxicity of CNTs, since an extensive dissertation on strategies to limit this toxicity is available in our recent article [2], here, we reported only a collection of some important tactics suggested so far to lessen the possible dangerous outcomes that could come from an extensive exposure to CNTs, as shown in Table 10.

## 5. Conclusions, Preventive Behavior and Future Perspectives

Through this review, we have highlighted the double-sided impact that the use of CNTs in nanocomposites finalized to restore a healthy equilibrium by efficient environmental remediation could have. Paradoxically, while CNT-enriched nanocomposites demonstrated remarkably higher efficiency in the removal of pollutants from water, soil and air, they could represent a strong risk for the environment itself and living organisms’ health. According to the several laboratory and practical studies reviewed in this work, CNTs have demonstrated high versatility in removing different types of xenobiotics, including heavy metals, organic pollutants, dyes, phenols, pharmaceuticals, pesticides, etc., from water and air. Specifically, CNTs are capable of remediating the environment by adsorption, filtration and catalytic oxidative degradation. Additionally, CNT-based filters have demonstrated high efficiency in removing microorganisms and their toxins from water and high potential for water desalination applications. In summary, membranes enriched with CNTs outperform conventional water, GW and WW treatment technologies, possessing improved water permeability, selectivity, faster filtration capability and efficient antifouling characteristics. Chemical surface modifications and/or the combination of CNTs with metal oxide nanoparticles, natural and synthetic molecules, macromolecules, polymers and dendrimers have been experimented with to further enhance CNT-based nanocomposites’ efficiency in environmental remediation. CNT-based nanocomposites offer several advantages, including high adsorption efficiency, rapid kinetics and the possibility of regeneration for multiple cycles. Despite the notable recent progress made to address the major challenges associated with CNT utilization, such as aggregation, toxicity, regeneration, etc., further research is necessary to better optimize CNT-based nanocomposites’ performance in environment remediation, to improve their scalability and to explore innovative strategies for CNT regeneration. Also, although more eco-friendly and sustainable production methods, including the utilization of raw biomass as a carbon source, are being extensively explored, the manufacturing cost of CNTs is a residual hurdle that needs to be solved by finding cost-effective and safer methods for their synthesis. Finally, although several strategies have been developed to limit the toxic effects on humans and the environment possibly deriving from the extensive use of CNTs, their cytotoxicity, genotoxicity and cancerogenic effects still represent the most risky and worst side of their environmental application. It is of paramount importance to better clarify the level of noxiousness of CNTs to limit the large diffusion of their possible cytotoxic impact due to their environmental accumulation and long persistence in bodies. However, preventive actions should also be conducted. Biocompatibility assays using appropriate in vitro and in vivo models must be considered before the widespread use of CNTs. Assessing the specific risk of a precise CNT structure is critical for knowing the individual biological responses to different types of CNTs. Notably, while waiting for a more detailed regulation, which is urgently needed for these nonpareil nanomaterials, it is essential to adhere to existing regulations and guidelines on safe nanomaterial use. The responsible development and use of CNTs can only be ensured by ensuring compliance with regulatory standards. Workers who produce, handle and apply CNTs need adequate training and education that makes them aware of potential risks that could arise from the incorrect use of CNTs identified until now. The implementation of proper safety protocols could help minimize harmless exposure to CNTs. Monitoring programs for evaluating the dispersion and potential impact of CNTs in workplaces and surrounding environments could help in updating risk management strategies. Unfortunately, efficient standardized techniques for assessing the toxicity of these nanomaterials are still troubled by difficulties and restrictions. Permanent research and collaboration between scientists, engineers and regulators are pivotal for developing safe routines for CNT production and use in various industries.

## Data Availability

No new data were created for this manuscript.

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
