# Peer review of "Carbon-Nanotube-Based Nanocomposites in Environmental Remediation: An Overview of Typologies and Applications and an Analysis of Their Paradoxical Double-Sided Effects"

_jox, 2025, doi:10.3390/jox15030076_

Round 1

Reviewer 1 Report

Comments and Suggestions for Authors

The review article “The Paradoxical Double-Sided Effects of Carbon Nanotubes-Based Nanocomposites in Environmental Remediation” by Silvana Alfei and Guendalina Zuccari is devoted to the description of the structure, properties, methods of synthesis of carbon nanotubes and composites based on them. The article shows the main aspects of the use of carbon nanotubes for solving environmental problems, such as water purification, sorption of various substances, desalination, etc. A separate section in the review is devoted to toxicological studies of materials based on carbon nanotubes. The review is interesting and voluminous, it will be useful for a wide range of readers. The authors cite a large number of publications, both new and works that stand at the origins of nanotube research. The work is well worth publishing in Jo X (Journal of Xenobiotics).

- As for the shortcomings, it is worth noting that the review is too overloaded with material. But the authors tried to summarize it in tables, which is undoubtedly more convenient for reading.

- A question about the figures. Some figures are provided without references. These are original figures of the authors, or it is worthwhile to indicate their sources.

- Table S1. Main methods to synthesize carbon nanotubes (CNTs) does not include the method of aerosol synthesis of nanotubes. The table should be supplemented with the following data:

Aerosol CVD

CNTs are synthesized in the gas phase and deposited in the form of randomly oriented networks, ready for many applications including transparent conductors.

SWCNT thin films

Clean defectless SWCNTs with limited yield

References:

Nasibulin, A. G., Moisala, A., Brown, D. P, Jiang, H. and Kauppinen, E. I. (2005) A novel aerosol method for single-walled carbon nanotube synthesis, Chemical Physics Letters 402, 227-232.

Eldar M. Khabushev, Julia V. Kolodiazhnaia, Dmitry V. Krasnikov, Albert G. Nasibulin (2021) Activation of catalyst particles for single-walled carbon nanotube synthesis. Chemical Engineering Journal 413, 127475. (PDF)

Ilya V. Novikov, Dmitry V. Krasnikov, Il Hyun Lee, Ekaterina E. Agafonova, Svetlana I. Serebrennikova, Yeounggyu Lee, Sihyeok Kim, Jeong-Seok Nam, Vladislav A. Kondrashov, Jiye Han, Ignat I. Rakov, Albert G. Nasibulin, Il Jeon (2025) Aerosol CVD Carbon Nanotube Thin Films: From Synthesis to Advanced Applications: A Comprehensive Review. Advanced Materials, 2413777.

If a review is checked for plagiarism and is an original work, it may well be published with minor edits.

Author Response

The review article “The Paradoxical Double-Sided Effects of Carbon Nanotubes-Based Nanocomposites in Environmental Remediation” by Silvana Alfei and Guendalina Zuccari is devoted to the description of the structure, properties, methods of synthesis of carbon nanotubes and composites based on them. The article shows the main aspects of the use of carbon nanotubes for solving environmental problems, such as water purification, sorption of various substances, desalination, etc. A separate section in the review is devoted to toxicological studies of materials based on carbon nanotubes. The review is interesting and voluminous, it will be useful for a wide range of readers. The authors cite a large number of publications, both new and works that stand at the origins of nanotube research. The work is well worth publishing in Jo X (Journal of Xenobiotics).

- As for the shortcomings, it is worth noting that the review is too overloaded with material. But the authors tried to summarize it in tables, which is undoubtedly more convenient for reading.

We thank the Reviewer for having understand our strategy to use Tables, to collect the huge information existing on the application of carbon nanotubes for environmental remediation. Our intention was in fact, to provide readers with an all-round information, about the use of CNTs to remove pollutants from different environmental districts and employing different techniques, in a single work. This, for capturing the attention of experts from different research area and a wider range of readers. To this end, we have embraced such a broad topic, that only the use of Tables would have made the resulting work better readable. We could not then fail to bestow to these unparalleled materials a due background/introduction and to mention their possible toxicity and poor regulation issues.

- A question about the figures. Some figures are provided without references. These are original figures of the authors, or it is worthwhile to indicate their sources.

We thank the Reviewer for his/her request, that enabled us to better explain the origin of Figures without reference (Figure 5 and Figure 6). We assure the Reviewer and Jox Editors that Figure 5 and Figure 6, without reference, are original and produced by us.

- Table S1. Main methods to synthesize carbon nanotubes (CNTs) does not include the method of aerosol synthesis of nanotubes. The table should be supplemented with the following data:

Aerosol CVD

CNTs are synthesized in the gas phase and deposited in the form of randomly oriented networks, ready for many applications including transparent conductors.

SWCNT thin films

Clean defectless SWCNTs with limited yield

References:

Nasibulin, A. G., Moisala, A., Brown, D. P, Jiang, H. and Kauppinen, E. I. (2005) A novel aerosol method for single-walled carbon nanotube synthesis, Chemical Physics Letters 402, 227-232.

Eldar M. Khabushev, Julia V. Kolodiazhnaia, Dmitry V. Krasnikov, Albert G. Nasibulin (2021) Activation of catalyst particles for single-walled carbon nanotube synthesis. Chemical Engineering Journal 413, 127475. (PDF)

Ilya V. Novikov, Dmitry V. Krasnikov, Il Hyun Lee, Ekaterina E. Agafonova, Svetlana I. Serebrennikova, Yeounggyu Lee, Sihyeok Kim, Jeong-Seok Nam, Vladislav A. Kondrashov, Jiye Han, Ignat I. Rakov, Albert G. Nasibulin, Il Jeon (2025) Aerosol CVD Carbon Nanotube Thin Films: From Synthesis to Advanced Applications: A Comprehensive Review. Advanced Materials, 2413777.

We greatly appreciated the notable help given us by the Reviewer with this new information. Aerosol CVD method, with details furnished by the Reviewer, has been included in Table S1 with the related references.

If a review is checked for plagiarism and is an original work, it may well be published with minor edits.

We are not sure to have understand this comment and apologise in advance to the Reviewer for this. Anyway, we have double checked the manuscript to reduce, where possible, existing duplicates on indication of the Editorial Office. Anyway, we ask Editors and Reviewer to take in mind that some terms, such as those indicating materials, material characteristics, synthetic methods, environmental districts, etc, cannot be modified.

Reviewer 2 Report

Comments and Suggestions for Authors

The Paradoxical Double-Side Effects of Carbon Nanotubes Based Nanocomposites in Environmental Remediation” authored by Alfei and Zuccari covers a wide range of works concerning the use of CNTs and CNTs based nanocomposites for environmental remediation. The review is well organized, presenting a lot of aspects about CNTs and their application in adsorption processes. However, some questions should be addressed before publication in “Journal of Xenobiotics”:

  • The authors deeply analysed bibliography presenting a great number of works regarding the use of CNTs in adsorption processes, so much so that the large part of this review is fully dedicated to the description of this process. Finally, a short paragraph about the paradoxical toxicity of CNTs, which should be the aim of this review, is presented. Considering this organization, is the review title really exhaustive and in line with the scope of this review? Probably, a title of the form “Carbon Nanotubes Based Nanocomposites in Environmental Remediation: an overview on typologies, applications and analysis of double-side effects” could be more appropriate.
  • Despite the huge work done by the authors, some recent literature works about the use of CNTs membranes (in the form of buckypapers) for the removal (through adsorption or photocatalysis) of wastewater pollutants are missing.
  • Considering the length of the review and the multitude of aspects to be analysed, paragraph 1 “What are carbon nanotubes?” should be shortened. A brief introduction about differences among SWCNTs, DWCNTs and MWCNTs and synthesis methods is sufficient, since the role of CNTs as nanomaterials of great properties is a notable fact.
  • Considering that some existing reviews about CNTs application in adsorption processes can be found in literature, Table S12, Table S13, Table S14 should be introduced in the manuscript, to strengthen the real novelty of it.
  • Please uniform the acronym for carbon nanotubes. For example, line 38, correct “CTS” in “CNTs”. The same for the acronyms of single walled carbon nanotubes “SWCNTs” and multi walled carbon nanotubes “MWCNTs”.
  • Line 605. Correct “MWCANTs”.

Author Response

The Paradoxical Double-Side Effects of Carbon Nanotubes Based Nanocomposites in Environmental Remediation” authored by Alfei and Zuccari covers a wide range of works concerning the use of CNTs and CNTs based nanocomposites for environmental remediation. The review is well organized, presenting a lot of aspects about CNTs and their application in adsorption processes. However, some questions should be addressed before publication in “Journal of Xenobiotics”:

  • The authors deeply analysed bibliography presenting a great number of works regarding the use of CNTs in adsorption processes, so much so that the large part of this review is fully dedicated to the description of this process. Finally, a short paragraph about the paradoxical toxicity of CNTs, which should be the aim of this review, is presented. Considering this organization, is the review title really exhaustive and in line with the scope of this review? Probably, a title of the form “Carbon Nanotubes Based Nanocomposites in Environmental Remediation: an overview on typologies, applications and analysis of double-side effects” could be more appropriate.

We thank a lot the Reviewer for this comment which we appreciated. We agree with the Reviewer, therefore the title has been changed on the Reviewer suggestion in “Carbon Nanotubes-Based Nanocomposites in Environmental Remediation: An Overview on Typologies, Applications and Analysis of Their Paradoxical Double-Side Effects”. Please, see lines 2-4.

  • Despite the huge work done by the authors, some recent literature works about the use of CNTs membranes (in the form of buckypapers) for the removal (through adsorption or photocatalysis) of wastewater pollutants are missing.

We are grateful to the Reviewer for his/her suggestion. The comment of the Reviewer has enabled us to correct a big omission. In this regard, an extensive part on buckypaper membranes have been added in the main text with additional two Tables and related references. Please, see lines 778-826.

  • Considering the length of the review and the multitude of aspects to be analysed, paragraph 1 “What are carbon nanotubes?” should be shortened. A brief introduction about differences among SWCNTs, DWCNTs and MWCNTs and synthesis methods is sufficient, since the role of CNTs as nanomaterials of great properties is a notable fact.

On suggestion of the Reviewer who we thank, paragraph 1 has been significantly shortened. Please, consider the revised sections 1, 1.1., 1.2. and 1.3.

  • Considering that some existing reviews about CNTs application in adsorption processes can be found in literature, Table S12, Table S13, Table S14 should be introduced in the manuscript, to strengthen the real novelty of it.

As asked, Table S12, S13 and S14 have been moved from Supplementary Materials to the main text where appear as Table 8, 9 and 10.

  • Please uniform the acronym for carbon nanotubes. For example, line 38, correct “CTS” in “CNTs”. The same for the acronyms of single walled carbon nanotubes “SWCNTs” and multi walled carbon nanotubes “MWCNTs”.

We thank the Reviewer for his/her comment. The acronyms have been uniformed.

  • Line 605. Correct “MWCANTs”.

Done.

Reviewer 3 Report

Comments and Suggestions for Authors

This review summarizes the studies carbon nanotubes for applications in the field of water purification and air purification. The authors also raise the concern about possible toxicity of the carbon nanotubes (for human and environment). The review of interest and can be considered for publication.  The part related to carbon nanotube toxicity must include more recent studies from 2020 to 2024. Many studies cited are probably outdated. I recommend major revision. Here are the detailed comments:

  • Line 408, the authors mention that by-product of dye degradation are toxic and carcinogenic. Why researcher study their degradation using photocatalysis if the by-products are toxic? The authors must clarify what kind of by-product are toxic and why. Part 2.1.2 mention that by-*product are toxics, but part 2.1.3 describe the advantages of degrading dyes…
  • Line 457, the authors mention issues related to the use of semi-conductor use for photocatalysis. What are these issues? Should we give up the use of semiconductor for catalysis? The authors must be precise and argue. Many research groups may not agree with that sentence, and critics appears to high considering the number of research papers published in the field. Rapid research shows that now research consider films of nanoparticles to recover them easily (doi.org/10.1007/s11356-024-35785-3).
  • Line 298-300 “GW is of paramount importance for …” must be clarified. GW is usually not used for these activities.
  • Line 493, it is not clear how CNT promote the adsorption of hydroxyl groups on MONP. The authors must clarify.
  • Reference 105 is unclear and cannot be found.
  • Line 405 the color is not the main issue. Other more serious issues must be mentioned.
  • The references related to the toxicity studies are too old, they can be kept, but more recent studies must be cited. For example, refernce 328 (line 954) was published in 2013 and reference 329 (line 955) was published in 2008.
  • ALD has been also use to combine vanadium oxide with CNT for gas sensor application (https://doi.org/10.1039/B821555C)
  • If air purification is cost-effective (line 823) and efficient, the author must mention an applied technology. The authors must add example, the technology has been developed for years.
Comments on the Quality of English Language

The English needs to be carefully check, it contains typos and inappropriate wording for a scientific review.

  • Line 249, recovering is inappropriate. Line 404 “ruinously” is inappropriate. Many other words are also inappropriate.
  • Line 596, form must be corrected. Line 605 what is MWCANT ?

Author Response

This review summarizes the studies carbon nanotubes for applications in the field of water purification and air purification. The authors also raise the concern about possible toxicity of the carbon nanotubes (for human and environment). The review of interest and can be considered for publication.  The part related to carbon nanotube toxicity must include more recent studies from 2020 to 2024. Many studies cited are probably outdated. I recommend major revision. Here are the detailed comments:

  • Line 408, the authors mention that by-product of dye degradation are toxic and carcinogenic. Why researcher study their degradation using photocatalysis if the by-products are toxic? The authors must clarify what kind of by-product are toxic and why. Part 2.1.2 mention that by-*product are toxics, but part 2.1.3 describe the advantages of degrading dyes…

We thank a lot the Reviewer for his/her comment, which enabled us to be clearer in the discussion on this part. As reported, the photocatalytic degradation of dyes and other organic pollutants is mainly based on advanced oxidative processes (AOPs), which have many advantages over other degradative techniques. The photocatalytic process, depending on conditions, comprise of domain steps in degradation pathways, including demethylation, ring shortening, ring opening, hydroxylation, addition of NO2 radicals and finally, mineralisation (https://doi.org/10.1016/j.greeac.2025.100230, https://doi.org/10.1016/j.eti.2021.102198). Practically, under the conditions of AOPs-based photocatalytic degradation, the pollutant molecules are mineralized in inorganic simple and no longer toxic molecules, rather than being transformed in other toxic byproducts, as occurs during simple not catalytic photolytic degradation. This occurs for a wide range of pollutants including dyes, other organic molecules, inorganic molecules, and pathogens. The AOPs do not create secondary pollutants, but small inorganic ions are produced as final products. Please, see also https://doi.org/10.1007/s11356-021-16389-7. The toxic and cancerogenic products we intended in the text, are those which can derive by the simple not-complete photolysis of dyes. On the contrary, such products do not survive when the photocatalytic degradation is used, especially if supported by CNTs. Anyway, for more clarity, these explanations have been included in the text. The schematic pathway of methylene blue (MB) degradation by an UV/nitrate photocatalytic process has been provided, and toxic products have been specified. Please, see lines 441-476.

  • Line 457, the authors mention issues related to the use of semi-conductor use for photocatalysis. What are these issues? Should we give up the use of semiconductor for catalysis? The authors must be precise and argue. Many research groups may not agree with that sentence, and critics appears to high considering the number of research papers published in the field. Rapid research shows that now research consider films of nanoparticles to recover them easily (doi.org/10.1007/s11356-024-35785-3).

We thank the Reviewer for this comment. In this regard, we make kindly note to the Reviewer that we have not criticized the potential of semiconductors in catalysis and far be it from us, to think of not using semiconductors in catalysis anymore. We have instead highlighted the limitations of metal oxide nanoparticles as catalysts in photocatalytic degradation processes, if used alone. These limitations, which the Reviewer would wish to be indicated, were already present in the original version of our manuscript in a dedicated paragraph (CNTs-Assisted Photocatalytic Degradation of Organic Pollutants: The proposed Mechanism). But a hint to other problems was already present earlier in the work at lines 514-420. Moreover, we thank the Reviewer for having provided us the relevant article on the use of mixtures of Cu NPs both freestanding and immobilized on glass surfaces to degrade dyes and better recovering the used catalysts. Anyway, we make kindly note to the Reviewer, that in their work, Authors have evidenced the same issues of semiconductors evidenced by us, if these semiconductors are used alone in photocatalysis.

  • Line 298-300 “GW is of paramount importance for …” must be clarified. GW is usually not used for these activities.

We appreciate the comment of Reviewer which enabled us to delve deeper into this part. Additional uses of GW have been added. Please, see lines 315-324.

  • Line 493, it is not clear how CNT promote the adsorption of hydroxyl groups on MONP. The authors must clarify.

As asked, more explanations on why and how CNTs promote the adsorption of more OH groups on MONPs have been added in the main text with related reference. Please, see lines 552-557.

  • Reference 105 is unclear and cannot be found.

We thank the Reviewer for this report. Reference 105 has been added with further details and corrected. Now it is clearer and can be found.

  • Line 405 the color is not the main issue. Other more serious issues must be mentioned.

On suggestion of Reviewer, we have double checked this part on the impact of dyes on water and we have found that other issues in addition to the colour were already present in the original version of our manuscript. However, to satisfy the Reviewer, other problems related to the presence of these pollutants in water were added. Please, see lines 428-441.

  • The references related to the toxicity studies are too old, they can be kept, but more recent studies must be cited. For example, refernce 328 (line 954) was published in 2013 and reference 329 (line 955) was published in 2008.

Dear Reviewer, thanks for your help. The references indicated have been updated. They are changed with Awasthi et al., 2024 and Jamwal, 2021. Additionally, three additional Tables (Tables 8, 9 and 10) have been included in the main text, containing more recent studies on CNTs toxicity.

  • ALD has been also use to combine vanadium oxide with CNT for gas sensor application (https://doi.org/10.1039/B821555C)

The paper suggested by the Reviewer has been considered and this study has been used to improve and complete Section 2.2. Please, see lines 917-922.

  • If air purification is cost-effective (line 823) and efficient, the author must mention an applied technology. The authors must add example, the technology has been developed for years.

As asked, applied technologies have been mentioned in the main text together with the main mechanisms governing air filtration by conventional filters and CNTs-based ones. Please see lines 937-951.

Comments on the Quality of English Language

The English needs to be carefully check, it contains typos and inappropriate wording for a scientific review.

  • Line 249, recovering is inappropriate. Line 404 “ruinously” is inappropriate. Many other words are also inappropriate.
  • Line 596, form must be corrected. Line 605 what is MWCANT?

We thank the Reviewer for his/her help in suggesting points were typos, grammatical errors and inappropriate words were present. The signalled issues have been corrected. In addition, our work has been revised also by Professor Deirdre Kantz, expert English mother tongue, who is our colleague, teaching English in University of Genoa and Pavia.

Round 2

Reviewer 3 Report

Comments and Suggestions for Authors

The paper is well written. The presented review fits the scopes and requirements for being published in Journal of Xenobiotics. I have had a careful look at the authors’ responses and at the modifications performed in the manuscript. The authors performed all the necessary changes to improve their paper. I am satisfied of the changes made, and in my opinion the paper can now be accepted for publication in Journal of Xenobiotics.